# Interface Strengthening of PS/aPA Polymer Blend Nanocomposites via In Situ Compatibilization: Enhancement of Electrical and Rheological Properties

**DOI:** 10.3390/ma14174813

**Published:** 2021-08-25

**Authors:** Lilian Azubuike, Uttandaraman Sundararaj

**Affiliations:** Department of Chemical and Petroleum Engineering, University of Calgary, 2500 University Drive NW, Calgary, AB T2N 1N4, Canada; lilian.azubuike@ucalgary.ca

**Keywords:** polymer blend nanocomposites, in situ compatibilization, interface, polystyrene, amorphous nylon, carbon nanotube CNT

## Abstract

The process of strengthening interfaces in polymer blend nanocomposites (PBNs) has been studied extensively, however a corresponding significant enhancement in the electrical and rheological properties is not always achieved. In this work, we exploit the chemical reaction between polystyrene maleic anhydride and the amine group in nylon (polyamide) to achieve an in-situ compatibilization during melt processing. Herein, nanocomposites were made by systematically adding polystyrene maleic anhydride (PSMA) at different compositions (1–10 vol%) in a two-step mixing sequence to a Polystyrene (PS)/Polyamide (aPA) blend with constant composition ratio of 25:75 (PS + PSMA:aPA) and 1.5 vol% carbon nanotube (CNT) loading. The order of addition of the individual components was varied in two-step mixing procedure to investigate the effect of mixing order on morphology and consequently, on the final properties. The electrical and rheological properties of these multiphase nanocomposite materials were investigated. The optical microscope images show that for PS/aPA systems, CNTs preferred the matrix phase aPA, which is the thermodynamically favorable phase according to the wettability parameter calculated using Young’s equation. However, aPA’s great affinity for CNT adversely influenced the electrical properties of our blend. Adding PSMA to PS/aPA changed the structure of the droplet phase significantly. At 1.5 vol% CNT, a more regular and even distribution of the droplet domains was observed, and this produced a better framework to create more CNT networks in the matrix, resulting in a higher conductivity. For example, with only 1.5 vol% CNT in the PBN, at 3 vol% PSMA, the conductivity was 7.4 × 10^−2^ S/m, which was three and a half orders of magnitude higher than that seen for non-reactive PS/aPA/CNT PBN. The mechanism for the enhanced conductive network formation is delineated and the improved rheological properties due to the interfacial reaction is presented.

## 1. Introduction

Polymer blends for the most part are thermodynamically immiscible, however the combination of different polymers creates a significant advantage for final polymer properties [1]. The morphology and the interfaces between the micro and nano phases influence the macroscopic properties of polymer blends [2] and thus, we can tailor the property profile of the resulting multiphase system. The performance of the blend depends greatly on the interfacial interactions, and normally for phase separated blends, the interfaces are usually weak, this can be attributed to the ability of few chains penetrating enough into the other phase to become entangled with chains on the other side of the interface to have a strong interfacial bond [3]. In polymer blends, filler addition has been shown to impact the general properties of the system [4,5]. For polymer blend nanocomposites, we need to strategically enhance the interfacial adhesion such that the composites’ mechanical properties are significantly improved. The fillers tend to migrate between the polymer phases during melt processing and depending on the wetting characteristics of the filler with the blend components, they can be localized in either of the phases or at the interphase. This phenomenon usually can change the size of the dispersed phase droplets when the filler enters the interphase [5,6]. This is accompanied by an enhancement in tensile, electrical and rheological properties. The final properties are also determined by the inherent characteristics of the nanofillers, but the filler properties can only be transferred to the matrix if there is sufficient adhesion between the phases [4,7,8]. 

Carbon nanotube (CNT) is an outstanding filler with much higher aspect ratio and smaller diameter compared to other one-dimensional fillers. Since its discovery by Ijima [9] in the 1990s, this nanofiller has shown remarkable applications and has been used widely as a reinforcement agent in immiscible polymer blends to improve their mechanical, electrical and thermal properties. The introduction of CNTs in immiscible polymer blends is one strategy to modify and enhance the compatibility of the blends [10]. For a blend with components having varying affinity for CNTs, the filler will migrate to the preferred (i.e., favourable) component, and thus we can achieve selective localization of nanofillers in the blend. Figure 1 shows different localizations for CNT in a polymer blend. In the simplified schematic, we can see that CNT can be fully in one polymer phase (left and middle image) or CNT can be localized at the interface (right image). Though CNT presents some reinforcing capabilities [11,12] in polymer materials, strengthening the interface in polymer blend and enhancing the interfacial adhesion usually requires the addition of a third component that can bond the interface via immiscible polymer molecules entangling with a copolymer on either side of the interface [13]. In this work, we employed in situ reactive compatibilization of maleic anhydride in a PS/aPA blend system. The graft copolymer was formed at the interface during melt processing and thus, the copolymer was pinned at the interface, driving and trapping the nanofiller in strategic locations to achieve better conductivity and improved mechanical properties.

Polymer blends filled with CNTs can be utilized in both small and large-scale industrial applications, such as electrical conductivity, EMI shielding, charge storage and other mechanical applications like sporting goods. One of the main goals of this work was to enhance EMI shielding properties. EMI is unwanted radiation emitted from electronic devices or electrical circuits, that adversely affects other electronics and electrical devices; and this radiation is even dangerous to human health. Thus, there is a need to protect against these waves. Polymeric materials filled with CNTs can be suitable shields because of their characteristics, such as lightweight, non-corrosive and ease of processing of polymers compared to other conductive materials like metals. To obtain sufficient EMI shielding using CNTs requires the concentration of the nanofillers to reach or exceed the percolation threshold [14]. Therefore, to achieve an optimum network structure in the polymer blend nanocomposite during melt processing it is important to achieve an optimum dispersion within the matrix [15]. By addition of diblock, triblock and graft copolymer, the localization of CNT in a polystyrene/polypropylene (PS/PP) immiscible polymer blend was altered [16]. Many studies have shown that CNTs localization in polymer blends are primarily controlled by two major factors, thermodynamic [3,16,17] and kinetic [16,18] factors and these can be used to tune the final electrical properties of the multiphasic system.

In our work, we study two polymers polystyrene (PS) and amorphous nylon (aPA) with very dissimilar affinity for CNT. Since these two polymers are thermodynamically immiscible, the selective localization occurs predominantly due to thermodynamic factors. Hence, considering the analysis by Sumita et al. [19], using the interfacial tension between the polymers and between each polymer and the solid filler, we can predict in which phase the filler will localize. If the interfacial tension between one of the polymers and solid filler is greater than the sum of all other interactions in the system, the solid filler will localize in the other polymer phase. However, if the interfacial tension between each of the polymers and the solid filler is less than the sum of all the interactions in the system, the solid filler is predicted to localize at the interface. It should be noted that the selective localization of CNTs at the interphase has remained difficult due to factors like polymer viscosity, filler compatibility and transport behaviour [20]. Here, it is important to distinguish between interface typically used in small molecule studies and “interphase” in polymer studies, particularly when copolymers and filler are located between the phases. It is well known that the region between two immiscible polymers is quite broad, particularly if a compatibilizer like a copolymer is added to enhance interactions between the two phases. Then, the in-between region acts almost like an additional phase rather than a sharp interface. Therefore, in this work, we will use the term interphase over interface because in many cases we are using reactive compatibilization to form graft copolymer at the interface causing it to broaden and become an interphase.

In most studies, selective localization of nanofillers in the continuous phase and or in the interphase of an immiscible polymer blend has been denoted as an easy way to increase the electrical conductivity in a polymer blend [18,21,22]. The capability to connect the network of nanofillers is hindered by a lack of dispersion of the filler in the blend system, and here, since aPA has very strong affinity for CNTs, by encapsulation of CNTs by aPA molecules. Polymer encapsulation is detrimental to electrical properties since CNTs cannot connect with each other or at the very least be close enough to each other (~2.5 nm) to allow for electron tunneling [20,23]. To minimize CNT encapsulation by nylon, we introduced a reactive polystyrene maleic anhydride (PSMA), to take advantage of the chemical reaction between maleic anhydride and amine groups in aPA (see Figure 2). The reactive compatibilization is achieved in-situ during melt processing of the polymer blend.

The rheological properties are also imparted as a result of the changes in the microstructure of the system, due to multi-interconnectivity of the nanofillers at the interphase resulting from the graft copolymer formed in situ and strong particle-particle interactions [22]. The storage modulus G’ usually exhibits a strong plateau behaviour at low frequency in networked polymer nanocomposites; hence, to investigate the microstructure of the blend system, we obtained small amplitude oscillatory shear (SAOS) linear viscoelastic properties, namely the storage modulus and the loss modulus as a function of frequency.

In this work, three mixing orders were utilized, for 25:75 PS/aPA blend composition with five different concentrations of styrene maleic anhydride (PSMA composition was 86 wt.% styrene and 14 wt.% MA). We studied the selective localization of CNTs, blend morphology of PS:aPA and the electrical and rheological properties of the PBN. The study suggests that a better electrical conductivity and EMI shielding performance can be obtained when the CNTs are first mixed in the minor phase and minimizing their migration into the thermodynamically preferred major phase by controlling the timing and extent of the amine-maleic anhydride reaction at the interphase. The dielectric permittivity of the blend composite was also measured and reported.

## 2. Materials and Methods

### 2.1. Materials and Composite Preparation

Polystyrene (PS) Styron 666D, (ρ = 1.04 g cm^−3^) was obtained from America Styrenics LLC (The Woodlands, TX, USA), Polyamide (aPA), HTN 503 (ρ = 1.14 g cm^−3^) was purchased from Dupont™ (Wilmington, DE, USA), and Zytel^®^, and polystyrene maleic anhydride (PSMA) Dylark^®^ 332, SMA-14 was acquired from Nova Chemicals (Calgary, AB, Canada). The multi-walled carbon nanotubes (MWCNTs) we used in our study were Nanocyl™ NC 7000, purchased from Nanocyl S.A (Sambreville, Belgium). The MWCNT was synthesized using catalytic vapor deposition (CVD) method with the following specifications, average diameter of 9.5 nm, 1.5 µm length and surface area range of 250–300 m^2^/g.

In all cases the PS-PSMA was the minor phase (lower concentration) and aPA was the major phase (higher concentration). In all PBNs, the PS-PSMA:aPA ratio was 25:75 by volume and the CNT was added at 0.5–3.0 vol% on top of that. The pellets of 25:75/PS:aPA were dried in a vacuum oven at 80 °C for 12 h. The polymer blend with 1.5 vol% CNTs were processed via the mixing orders (MOs) shown in Table 1 (i.e., we varied the orders of addition of the components). The table shows two mixing steps, where the first mixing step was carried out for 5 min and the subsequent mixing step occurred for an additional 10 min. The composites were prepared using Alberta Polymer Asymmetric Minimixer (APAM) [24] at 200 rpm and 240 °C, the volumetric percentage of all the constituents was calculated and the total volume of 2 cm^3^ used in the mini mixer. The processed composites were subsequently quenched in liquid nitrogen to stabilize the morphology to study the effect of melt mixing on CNT dispersion and localization. Thereafter, compression molding was performed using a Carver compression molder (Carver Inc., Wabash, IN, USA) at 240 °C for 10 min under 40 MPa pressure. The samples were molded into a rectangular shape with dimensions 11.0 × 22.0 × 1.0 mm^3^ to obtain samples for electrical conductivity and EMI shielding measurements.

### 2.2. Blend Characterization

The final morphology of the processed PBN was studied using optical microscopy (OM) and transmission electron microscopy (TEM). In preparing the samples for imaging, 1µm film thickness were cut from the molded samples at room temperature using a Leica^®^ Biosystems EM UC6 (Leica Biosystems^TM^, Nussloch, Germany) ultramicrotome. The images were then captured using an Olympus^®^ BX60 optical microscope (Olympus Inc, Tokyo, Japan) connected to an Olympus DP80 camera. For TEM, thin sections (90 nm) were cut using the ultramicrotome (Leica EM UC6) equipped with glass knife, at a cutting speed of 0.4 mm/s at room temperature, the images were obtained using Hitachi HT7500, equipped with Olympus SIS Mega View II, 1.35 MB digital camera.

The DC conductivity measurement of the PBN was obtained from a Loresta GP resistivity meter (MCP-T610 model, Mitsubishi Chemical Co., Tokyo, Japan), this meter is connected to a four pin ESP probe (MCP, TP08P Model, Mitsubishi Chemical Co.). The electrical conductivity measurements were performed at an applied voltage of 90 V, and the reported results are averaged over three samples.

EMI properties were investigated with a wave guide network analyzer (ENA Model 5071C, Agilent Technologies, Santa Clara, CA, USA) within X-band frequency range of 8.2 to 12.4 GHz. Rectangular molded samples were placed between standard wave guides (WR-90). The S-parameters generated in the mentioned frequency range were used to calculate the EMI Shielding effectiveness, real and imaginary permittivity.

The rheological properties were measured using Anton-Paar MCR 302 Rheometer, with a 25 mm diameter parallel-plate geometry and gap size of 0.5 mm. The experiments were performed at a constant temperature of 240 °C. Strain amplitude sweep measurement was performed on all our systems to establish the linear viscoelastic region (LVR) for each sample. This examination was done over a range of applied strain from 0.1% to 1000% at an angular frequency of 1 rad/s. It was confirmed minimum strain of 0.1% was small enough to keep the deformation of our samples in the linear viscoelastic region. This Linear region is highly sensitive to any change in the microstructure of the polymer system.

## 3. Results and Discussion

### 3.1. Blend Morphology and CNTs Localization

Understanding the effect of the different processing conditions towards morphology evolution is critical to design PBN. In particular, the localization of CNTs plays an important role in final electrical, thermal and mechanical properties. The strength of the interfacial interaction amongst the constituent components of the blend nanocomposites also determines the blend structure [25]. The transmission electron micrographs in Figure 3 shows a representation of the CNTs dispersion and localization in the blend system under study and the effect of reactive compatibilization. The nanofillers can be seen to selectively localize in the major phase, and at the same time at the interphase. The localization of CNT at the interphase is due to the reaction of the chemical components of the major phase aPA and the compatibilizer PSMA achieved in-situ during processing. The reaction timing is optimized via the mixing order. In this case, the PSMA and aPA added after the CNT is well dispersed inside PS first (MO III).

Results from optical microscopy (OM) are shown in Figure 4, which displays the non-reactive PS/aPA/25:75 blend nanocomposite filled with 1.5 vol% CNT (i.e., without PSMA) prepared using the three mixing orders described in Table 1. All mixing orders led to the selective localization of CNT in the matrix phase as shown by OM images. The darker matrix phase represents the major phase (aPA), while the lighter dispersed phase is the minor phase (PS). For MO-I (Figure 4a and Figure 5a), it was clearly observed that CNTs were selectively localized in aPA, while for MO-II (Figure 4b and Figure 5b) and MO-III (Figure 4c and Figure 5c) CNTs were seen both in aPA matrix and at the interphase. In MO-II and MO-III, CNT was first introduced into the minor phase during melt processing; therefore, some CNTs as they were in the process of migrating to the thermodynamically favourable phase, remained at the interphase. In addition, with the addition of 3 vol% PSMA (Figure 5), the geometry of the droplet phase was significantly different for the three mixing orders. In MO-I, the droplet was slightly elongated and cut across the major phase with some CNT agglomerates. In MO-II, droplets were stretched along with the CNTs allowing for CNT network structure. Finally, in MO-III, the droplets exhibited a star-like structure with thin rays extending from the domains that connected the minor phases together and allowed for CNTs to form a conductive network.

The selective localization of CNTs in the aPA phase in our system was almost completely driven by thermodynamic factors due to the strong affinity between CNT and aPA; and this did not change even with the introduction of PSMA. The optical micrographs in Figure 6, Figure 7 and Figure 8 show the effects of addition of PSMA at different CNT concentrations (0.5 vol%–3 vol%), and illustrate the impact of the different mixing sequences on morphology. In MO-I, in the absence of PSMA the micrographs in Figure 6a,e,i show irregular shape of the droplet phase, and this irregularity increased with increase in CNT concentration. In system with 3 vol% CNT (Figure 6i), the dispersed phase domains were elongated, presumably due to droplet coalescence. Coalescence could be attributed to the absence of the compatibilizer and significant localization of CNT in the matrix phase. It has been shown for polymer blends that compatibilizer significantly impedes droplet coalescence [26] and that addition of CNT can enhance coalescence [27]. For MO-I, introduction of PSMA changed the droplet geometry to be smaller and more elongated. We observed a similar trend for 1.5 vol% CNT and 3 vol% PSMA for all mixing orders (Figure 6g, Figure 7g and Figure 8g), in which the minor phases were very stretched and cut across the major phase. This blend morphology tended to create a CNT network structure, consequently enhancing the electrical conductivity and electromagnetic interference shielding properties.

### 3.2. Electrical Conductivity and Electromagnetic Interference Shielding

To achieve outstanding electrical conductivity in an immiscible polymer blend system, it is critical to attain optimum filler distribution and dispersion during melt processing. In this regard, CNT localization should be tuned with respect to our targeted property—conductivity [28]. Conductivity can be achieved in polymer systems, when the conductive filler connects to form a network structure, which is known as percolation threshold [29,30]. The processing protocol (i.e., the order of addition of components) was the major factor utilized in this study to control filler migration. The second most important factor was introducing a compatibilizer during melt processing to accomplish in-situ compatibilization. The idea was to form copolymers via reactive polymer blending to form a larger and chemically linked interphase that would “trap” CNTs at the interphase or within the thermodynamically unfavorable phase. Thus, this in-situ compatibilization was used to achieve localization of the CNT nanofillers at the interphase to achieve significant increase in the electrical conductivity of our blend system at 1.5 vol% CNT composition. The electrical conductivity was measured for the nanocomposites 25:75/PS:aPA containing 1.5 vol% CNT, for 1, 3, 5 and 10 vol% PSMA for the three different mixing orders. In Figure 9a we plot the DC conductivity against the PSMA concentration for processed PBNs. It can be clearly seen that the conductivity at each mixing order changed with the increase in PSMA, and that there seemed to be an optimum PSMA concentration. The two major peak values of electrical conductivity were obtained for blend systems MO-III with 1 vol% PSMA and MO-II with 3 vol% PSMA.

The aPA component forms the matrix phase and had stronger affinity for CNT. Therefore, the nanofillers were drawn away from PS towards this major phase, in both MO-III and MO-II, even though the aPA was introduced later during melt processing. So, CNT migration from the minor PS phase to the major aPA phase resulted in the nanofillers being selectively localized at the interphase, and hence, we could achieve a high conductivity. The highest conductivity obtained was for the MO-II at 3 vol% PSMA. This can be attributed to the amount of graft copolymer formed during the in situ reactive compatibilization, resulting in more grafting and crosslinking at the interface. Consequently, we achieved a more connected network structure.

The EMI shielding effectiveness of the blend system prepared via melt processing technique, was calculated over X-band frequency, as this frequency range is used predominantly for most of the commercial applications [14]. Shielding effectiveness describes the material’s ability to attenuate electromagnetic interference. The different shielding mechanisms include reflection, absorption, and multiple reflections. For polymer nanocomposite materials, the major shielding contribution is via absorption [27,28], hence the microstructure, the nanofillers distribution, dispersion and connectivity within the system all contribute to the EMI shielding. Figure 9b shows the EMI shielding effectiveness (SE) at different amounts of PSMA for different mixing orders. Figure 10 displays the total shielding effectiveness (SE_T_) over the X-band frequency.

A significant increase of shielding effectiveness was observed by introducing reactive PSMA to the blend. Just like the electrical conductivity results, MO-III with 1 vol% PSMA and MO-II with 3 vol% PSMA gave the highest shielding effectiveness, 10.27 dB and 9.02 dB, respectively. This increase in EMI can be attributed to the state and the location of the in situ formed graft copolymer (aPA-g-SMA). Having introduced CNT first in the PS minor phase for both mixing orders (MO-II and MO-III), and not allowing any reaction between the two reactive components until the second stage, when the aPA matrix phase was introduced into the system and started reacting with PSMA, the formed copolymer at the interphase could stop the migration of CNT fully to the matrix phase such that that some CNT remained at the interpase.

In this study, at a fixed blend composition was used and the mixing process was varied producing different states of connectivity [14] of the aPA matrix component, into which the CNTs tend to migrate. Therefore, the mixing order gave different blend percolation, which subsequently affected CNT percolation. In additon, in some cases, localization of CNT at the interphase was attained. This state of this blend/interphase connectivity would have facilitated absorption of the incident EM wave [14]. MO-III and MO-II using low PSMA concentrations show relatively similar EMI attenuation performance and higher values of shielding effectiveness compared to MO-I. This is mainly as a result of segregated localization of CNT, amount of PSMA-aPA copolymer formed and the time given to form the copolymer as the different components are introduced at different times in the three mixing orders.

Though the two polymers are thermodynamically immiscible, each has different affinity for the nanofillers. aPA has a much higher affinity for CNTs than PS, however the aPA high affinity for CNTs has shown to be detrimental to establish network connection because they can coat the CNTs and not allow for the connections required for electrical conductivity. So systematically introducing CNT in the dispersed phase (PS) first with PSMA, and then adding aPA, can change where the CNTs locate as the CNTs will tend to migrate to the aPA matrix phase. With help of the PSMA, in which the maleic anhydride group reacts with amine group in aPA, the CNTs can be pinned at the interface if sufficient reaction occurs within the time that CNTs are migrating. In 1 and 3 vol% of PSMA of our system, the micrographs (Figure 6, Figure 7 and Figure 8) show stretched CNTs, hence making network connection possible and subsequently increasing the conductivity and rheological properties. Figure 11 below illustrates the concept for the different mixing orders.

### 3.3. Dielectric Properties: Real and Imaginary Permittivity

In this section, we investigated the impact of the in-situ reactive compatibilization of the blend at different composition of PSMA on the PBN dielectric properties for different mixing orders over the X-band frequency range. Figure 12 shows the real permittivity (ε′) and the imaginary permittivity (ε″) of the polymer blend nanocomposites for the three mixing orders with 1.5 vol% CNT versus the PSMA concentration. The measure of charge polarization in a material defined the ε′, which is the ability of the material to store charges. In nanocomposites, increase in real permittivity connected to the increase in nanocapacitors, which corresponds to high conductivity. This agreed with our conductivity results, as MO-II gave the highest dielectric constant of 15.28 at 3 vol% PSMA. With increase in PSMA content, there is significant reduction in the ε′ value. This could be attributed to a higher amount of in-situ reaction between PSMA and aPA, which led to a cross-linked system, preventing nanofillers from crossing the interphase and forming more agglomerates than a network structure. This is observed in Figure 12 as above 3 vol% PSMA, the ε′ value remained constant up to 10 vol% of PSMA.

Imaginary permittivity (ε″), on the other hand, represents the ohmic or dielectric loss in materials. In nanocomposites; this corresponds to microwave attenuation as result of the transport of charges in connected networks structure of CNT [14]. Consequently, electromagnetic wave attenuation in a material is function of both conduction loss and polarization loss [31] and is extremely detrimental for charge storage properties. Thus, we look for the lowest possible value for ε″. Often the dielectric loss tangent, tan δ = ε″/ε′, was used to characterize charge storage properties. MO-I with 1 vol% PSMA had the lowest tan δ value, indicating that it had the best charge storage capability. It should be noted that the highest EMI SE or conductivity usually did not translate to the best charge storage properties.

In our blend system, the results of EMI SE agreed with the conduction loss, i.e., for the ε″ value, MO-III with 1 vol% of PSMA gave the highest ε″ value and the highest SE_T_. This agreed with the study by Weir [32] that showed that high ε″ is needed for good electromagnetic interference shielding. Therefore, it could be established that the observed attenuation of the electromagnetic wave was introduced by a high dielectric loss in the system with 1 vol% PSMA.

### 3.4. Viscoelastic Properties

Figure 13 presents the SAOS rheological plots of the blend nanocomposites of the systems with 1 vol% and 3 vol% PSMA, which showed peak values for electrical conductivity and dielectric properties. The effect of the different mixing orders on the viscoelastic properties of the nanocomposites was evident. MO-III and MO-II showed significant increase in modulus over MO-I for 1 vol% and 3 vol% PSMA, respectively. This increase in the value of the plateau-storage and loss modulus could be attributed to the localization of the CNT at the interphase as the nanofillers migrated to the major phase, providing a better stress transfer across the phases, resulting in a significant increase in the storage modulus. The rheology results were in good agreement with conductivity results, which showed that a strong CNT network structure was formed in MO-III and MO-II at lower concentrations of PSMA (1 vol% and 3 vol%). Figure 13 shows that the network structure in MO-II breaks down at a lower critical strain amplitude (γ*) than MO-III and MO-I. This can be explained by the high degree of CNT connectivity with PSMA, which is more polar than PS, but this structure subsequently broke down with the introduction of the major phase aPA due to the CNT migration to the even more polar aPA and possibly due to more agglomeration.

## 4. Conclusions

The impact of polymer blend nanocomposite mixing processing (namely order of addition of components) on the development of blend microstructures, the degree of CNT connection, reactive in-situ compatibilization and the migration of the nanofillers into the thermodynamically preferred component was studied in this work for a fixed composition of PS/aPA blends containing different levels of CNTs and different concentrations of a reactive PSMA component. The morphology was studied using transmission optical microscopy and electron microscopy. The final properties of the PBNs, namely conductivity, EMI, dielectric and rheology, were analyzed and outstanding property values were obtained at particular mixing orders and PSMA concentrations. The timing of when the reaction occurred due to the mixing order and the level of reaction due to the compatibilizer concentration were used to explain the changes in properties. The blends prepared by melt mixing formed droplet in matrix microstructure, irrespective of the mixing order. However, the size and geometry of the drops varied significantly and typically became smaller and more elongated with increased PSMA concentration. Similarly, CNT addition also created a more irregular geometry and induced coalescence and connecting of dispersed phase domains.

The different mixing orders had a large significant effect on the connection between CNTs, and consequently, the final electrical properties. At 1 vol% and 3 vol% of the PSMA composition, we saw enhanced electrical and electromagnetic properties at 1.5 vol% of CNT. Using MO-II and 3.0 vol% PSMA, the electrical conductivity was 3.5 orders of magnitude higher than the same polymer blend nanocomposite using other mixing orders. The highest EMI SE of 10.27 dB, mostly due to absorption, was achieved in MO-III blends containing 1.5 vol% MWCNTs, and this PBN also showed high real and imaginary permittivity, and a significant increase in storage modulus in the viscoelastic rheology results. All the measured properties indicate that a better CNT network structure formed at MO-III and MO-II with lower PSMA compatibilizer concentrations. Thus, we saw a significant impact of in-situ compatibilization on electrical properties even at low concentration of the compatibilizers (1 vol%) by manipulating the process protocol.

## Figures and Tables

**Figure 1 materials-14-04813-f001:**
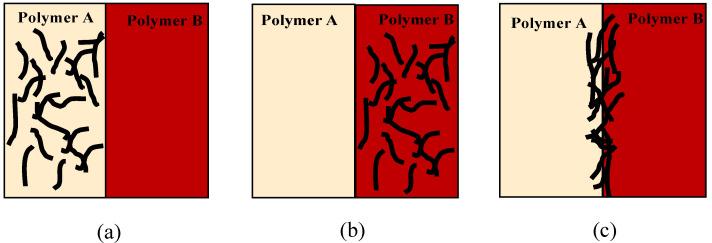
Schematic illustration of carbon nanotube (CNT) selective localization: (**a**) The CNT is completely within Polymer A, (**b**) CNT is fully within Polymer B and (**c**) CNT localizes at the interface.

**Figure 2 materials-14-04813-f002:**
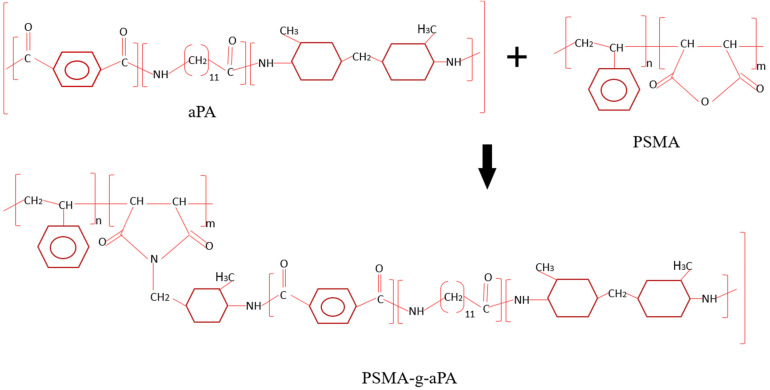
Chemical reaction between amine groups in aPA and malaeic anhydride groups in PSMA.

**Figure 3 materials-14-04813-f003:**
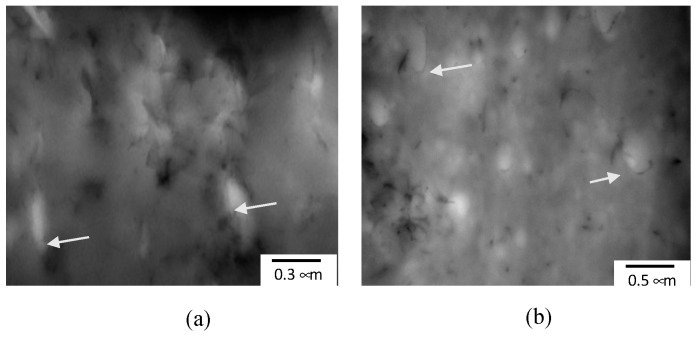
TEM of the PS/aPA (25:75) system, with 1.5 vol% CNT and 3 vol% PSMA (MO II) at different magnifications: (**a**) higher magnification, (**b**) lower magnification. It can be seen with PSMA addition, that the CNT selectively localize at the interface between PS and aPA. White arrows show CNT at the interface between PS drops and aPA matrix.

**Figure 4 materials-14-04813-f004:**
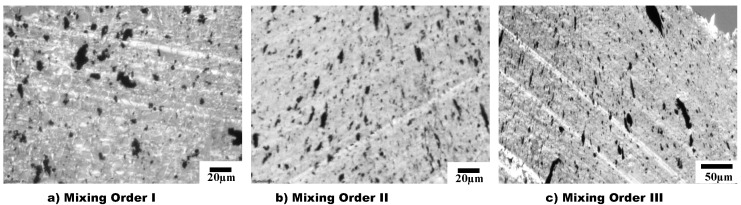
OM images of 1.5 vol% CNT in PS/aPA (25:75) blend nanocomposites for three different mixing orders (**a**) MO I, CNTs are selectively localized in the matrix; (**b**) MO II and (**c**) MO III, show stretched CNTs localized both at the matrix and at the interphase.

**Figure 5 materials-14-04813-f005:**
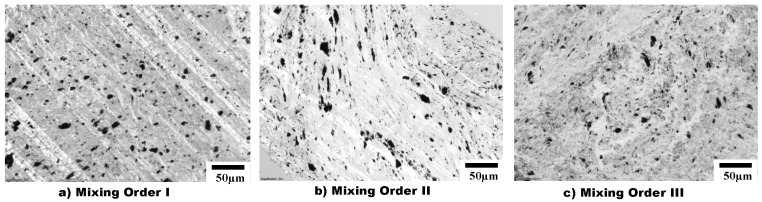
OM images of 1.5 vol% CNT in PS/PA66 (25:75) blend nanocomposites in three different mixing orders with 3 vol% PSMA. Introduction of PSMA results in diffrences in droplet geometry: (**a**) MO I, the droplets are slightly elongated, and CNTs form agglomerates. (**b**) MO II, the droplets are stretched along with CNTs, allowing for CNT network to form. (**c**) MO III, the droplets have a star-like structure that connects the minor phases, and creates a CNT network structure.

**Figure 6 materials-14-04813-f006:**
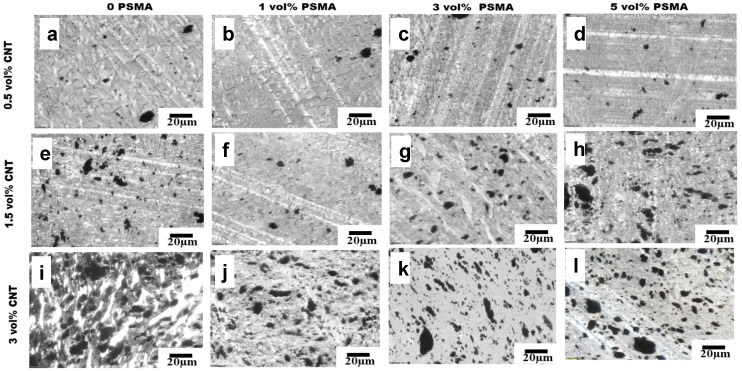
Mixing Order I. (**a**–**l**) Optical micrographs of PS-PSMA/aPA/CNT polymer blend nanocomposites. Going from left to right, the PSMA concentration in the blend increases and going from top to bottom, the CNT concentration in the PBN increases.

**Figure 7 materials-14-04813-f007:**
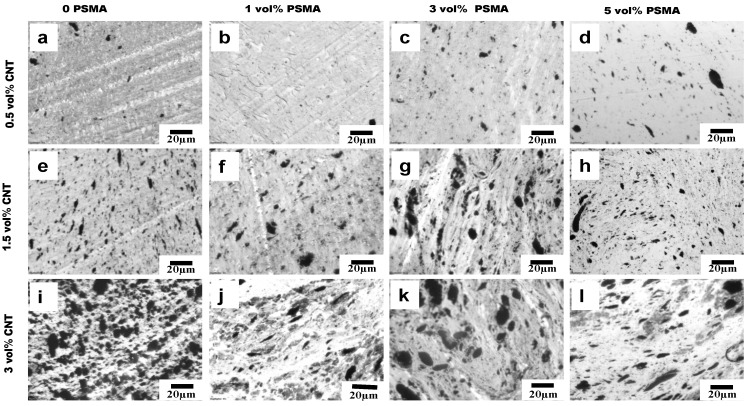
Mixing Order II. (**a**–**l**) Optical micrographs of PS-PSMA/aPA/CNT polymer blend nanocomposites. Going from left to right, the PSMA concentration in the blend increases and going from top to bottom, the CNT concentration in the PBN increases.

**Figure 8 materials-14-04813-f008:**
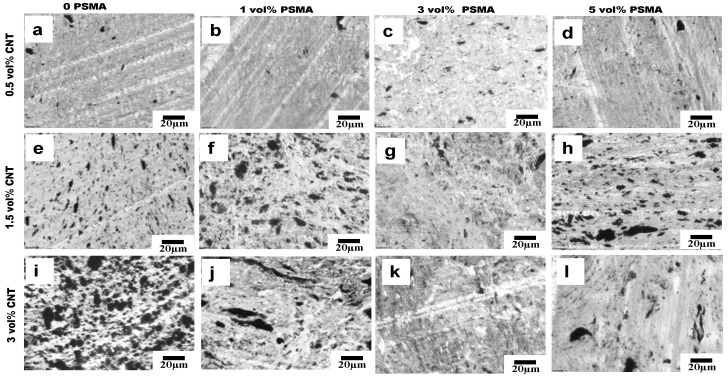
Mixing Order III. (**a**–**l**) Optical micrographs of PS-PSMA/aPA/CNT polymer blend nanocomposites. Going from left to right, the PSMA concentration in the blend increases and going from top to bottom, the CNT concentration in the PBN increases. Figure 8a,e,i are the same as Figure 7a,e,i because without any PSMA, MO-II and MO-III are identical.

**Figure 9 materials-14-04813-f009:**
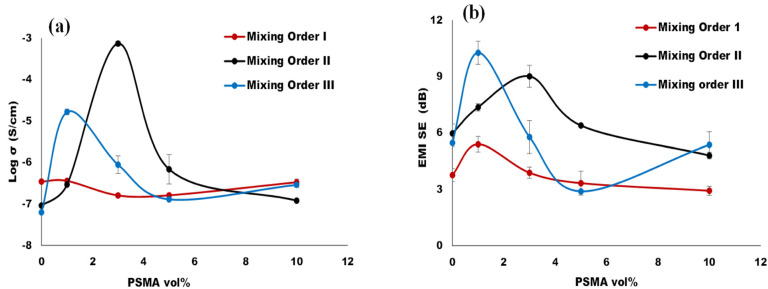
Properties obtained via different mixing orders. (**a**) Electrical conductivity and (**b**) EM SE of melt processed PS/aPA (25:79) as a function of PSMA content.

**Figure 10 materials-14-04813-f010:**
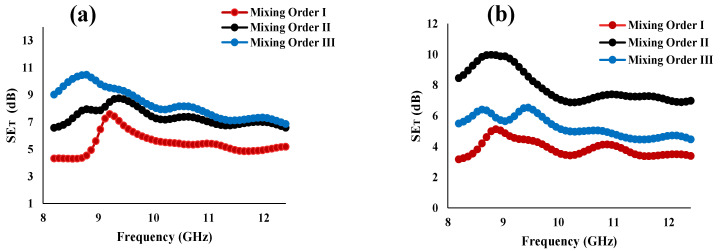
Total Shielding effectiveness (SE_T_) of melt processed PS/aPA6 (25:75)/5 vol% MWCNT (**a**) at l voI% PSMA and (**b**) 3 vol% PSMA. MO-II has best performance for 1 vol% PSMA but MO-III is best for 3 vo1% PSMA. MO-I consistently has lowest SE_T_.

**Figure 11 materials-14-04813-f011:**
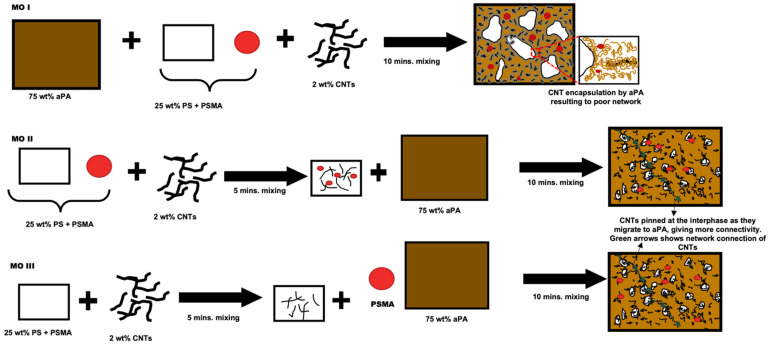
Schematic illustration of the mixing protocols.

**Figure 12 materials-14-04813-f012:**
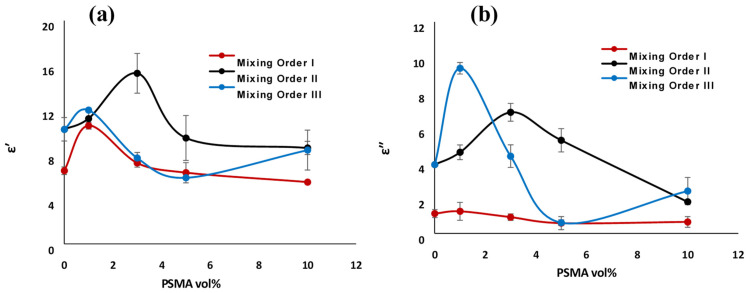
(**a**) Real permittivity and (**b**) imaginary permittivity of PS/aPA6/25:75 nanocomposites with 1.5 vol% CNT loading versus PSMA concentration.

**Figure 13 materials-14-04813-f013:**
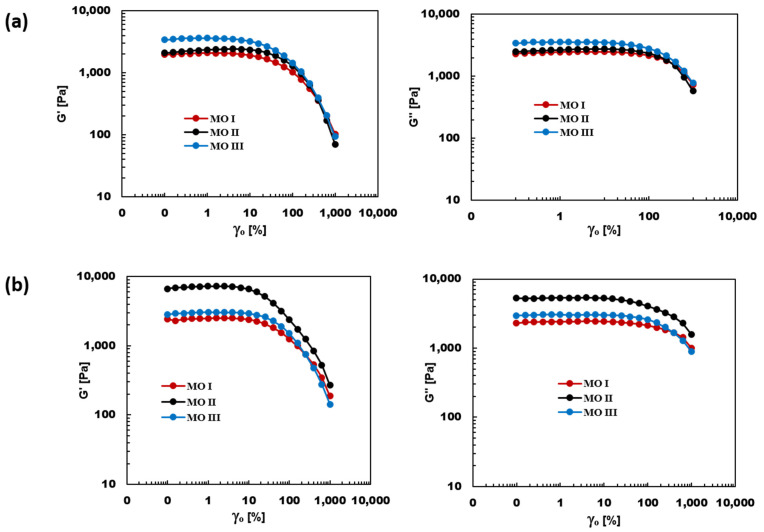
Storage (G′) and loss modulus (G″) of PS/aPA6 nanocomposites containing 1.5 vo1% CNT at (**a**) 1 vol% PSMA (**b**)3 vol% PSMA prepared using three different mixing orders.

**Table 1 materials-14-04813-t001:** Mixing order of addition.

Mixing Order	Step 1	Step 2
(MO)	t = 5 min	t = 10 min
I	PS + aPA	PSMA + CNT
II	PS + PSMA + CNT	aPA
III	PS + CNT	aPA + PSMA

## Data Availability

The data presented in this study are available on request from the corresponding author.

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
