# Peer review of "Interface Strengthening of PS/aPA Polymer Blend Nanocomposites via In Situ Compatibilization: Enhancement of Electrical and Rheological Properties"

_materials, 2021, doi:10.3390/ma14174813_

Round 1
Reviewer 1 Report
Too much is hang upon the term of compatibilization which is not sound
scientifically because it is not well defined, please do that.
Another issue would be on how do you achieve the sufficient
connectivity level for the electric charge?
Third, how do you interrelate electric and mechanical properties of the blend, and what is a clear
correlation thereof? Please do answer the three question raised above. Please do
consider to look for including to refs. a letter of 1996 on stochastic approach in biopolymeric
evolutions at Chem. Phys. Lett. 1996 by Gadomski A. wherein diffusion limited adsorption and domain ermegence
Reviewer 2 Report
The manuscript presents a series of complex polymer blend nanocomposites (PBNs) . In this sophisticated and complex system, electrical properties and rheological properties have been enhanced at the same time. Moreover, the authors found that by adding PSMA to PS/aPA changed the structure of the droplet phase significantly. This work is generally accurate and worthy of publication. But there are some concerns as follows, reviewer hope the authors could make amendments.
- PBNs involve four components: PS, aPA, PSMA and CNTs. I hope that the author can describe the overall picture of the system more clearly. Some parts of the current expression make it difficult for readers to understand.
- The chemical formula in Figure 2 is not formal, and a uniform bond length and bond angle could be used.
- The authors discussed in depth the impact of the three mixing order on performance, and the practicability (in industry) of the three blending methods (mixing order) should also be briefly discussed.
- The manuscript discussed morphology and rheology. It would be better if the tensile strength and elongation at break of the composite material could be given.
